# Invited Perspectives: "Small country, big challenges – Switzerland's hazard prevention research"

**Dorothea Wabbels[1], Gian Reto Bezzola[1]**

[1]Federal Office For The Environment (FOEN), Hazard Prevention Division, 3003 Berne, Switzerland

**Correspondence**: Dorothea Wabbels (dorothea.wabbels@bafu.admin.ch)

**Switzerland's situation at a glance.** Due to its geography and climate, densely populated Switzerland is often affected by water-related hazards such as surface runoff, floods, debris flows, landslides, rockfalls and avalanches. Strong earthquakes are rare, but, as history shows, they do occur and represent one of the biggest risks. Due to population growth, the expansion of settlements and infrastructure, the rise in mobility as well as the effects of climate change, risks ascribable to natural hazards
are increasing.

The hazard situation in alpine Switzerland is characterized by significant differences in altitude over short distances and relatively high precipitation volumes. In addition, the increase in temperature caused by climate change is higher, compared to the rest of Western Europe (FOEN 2020a).

**Natural hazards concern everyone in Switzerland.** Extreme natural events are a regular occurrence in Switzerland. For
instance the hail storms and floods of July 2021, the winter storm Burglind in 2018, the rock avalanche and debris flow in Bondo, canton Graubünden, in 2017, the floods of august 2007, august 2005 and may/june 1999 and the winter storm Lothar in 1999 affected large parts of the country and caused severe damages. However, the threat of hazards is not restricted to mountainous regions and along rivers and lakes. Almost every part of Switzerland is exposed to natural hazards and anyone can be affected. Data collected by the Federal Institute for Forest, Snow and Landscape Research (WSL 2021) on behalf of
the Federal Office for the Environment (FOEN) shows that in the past 49 years, four out of five Swiss communes have suffered damage due to flooding or debris flows (Fig. 1). Landslides affected two in five communes in the same period. Between 1972 and 2020 floods, debris flows, landslides and rockfall processes accounted for around 300 million Swiss francs of damage per year on average.

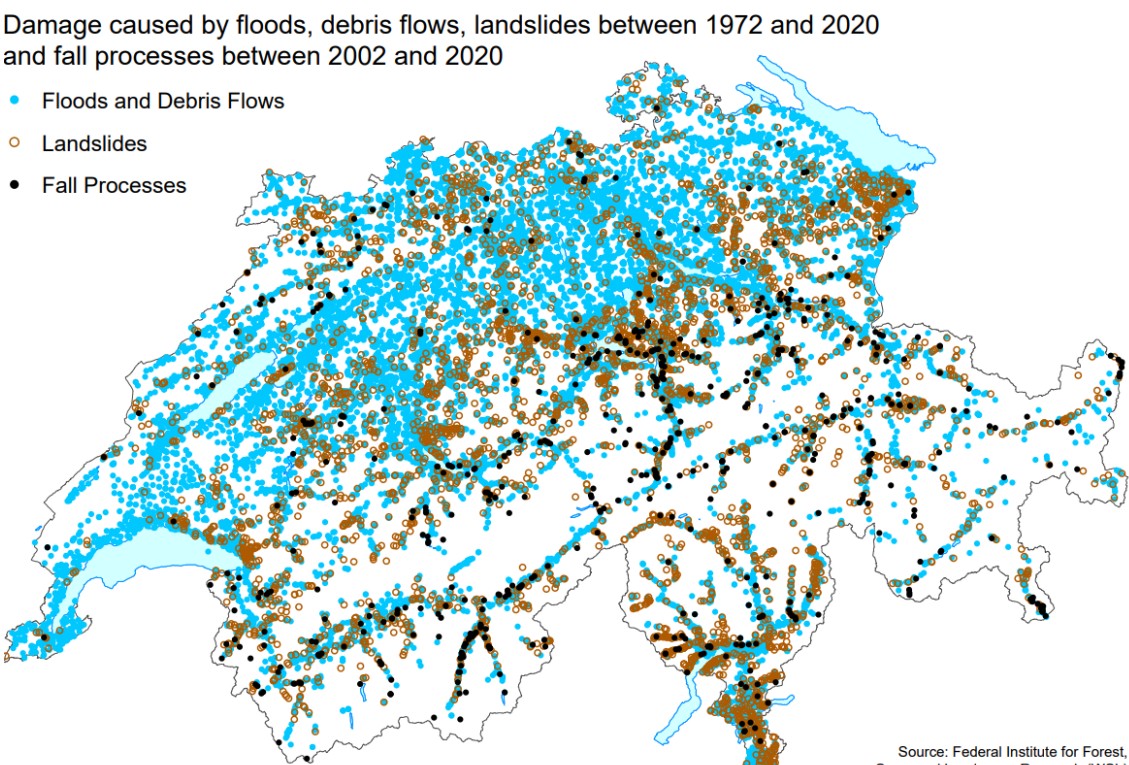

Damage caused by floods, debris flows, landslides between 1972 and 2020
and fall processes between 2002 and 2020

- Floods and Debris Flows
- Landslides
- Fall Processes

Source: Federal Institute for Forest,
Snow and Landscape Research (WSL)

**Figure 1: Damage caused by natural hazards in Switzerland. Federal Institute for Forest, Snow and Landscape Research (WSL).**

In Switzerland, there was a recognition and learning from a series of large events at the end of the 20th century that complete protection against natural hazards is impossible. Furthermore, over time it is apparent that structural measures alone are not sufficient. In terms of preventive measures, the priority shifted to spatial planning and land use respecting risks of natural hazards. The shift from simple hazard control to a risk culture required comprehensive data and information about the occurrence of hazards and the respective risks as well as the development of an integrated risk management. Integrated risk management as it is understood today in Switzerland, comprises the continuous and systematic identification, analysis and evaluation of risks, the planning and implementation of measures in response to the observed risks, based on the principles of cooperation, participation, mitigation and sustainability, and controlling the effectiveness of these measures. Integrated risk management is an approach where all natural hazards and all types of measures are considered, where all responsible actors participate in the planning and implementation of measures, and environmental, economic and social sustainability is envisaged (PLANAT 2014, 2018). Management of risks from natural hazards is a cross-sectional task, encompassing land use, spatial planning, water protection, environmental and civil protection by the government, cantons and communes, the private sector as well as insurance companies and individuals. These actors are involved in the process as a joint task, in accordance with the principle of subsidiarity.

**Small country, big challenges.** Switzerland has a long tradition concerning fundamental research as well as practice- and implementation-oriented research in hazard management and there is an important number of research institutions being active in this domain. The findings of these research activities will further improve the management of natural hazards and, taking into account environmental aspects, climate change and settlement densification. The «Research Concept Environment » (FOEN 2020b) describes the implementation of environmental policies in 18 research areas and thus the priority fields of

research from the perspective of FOEN in the near- and future-term. FOEN's focus on central research themes in hazard prevention of gravitational and tectonical hazards for 2021 – 2024 is listed below. The order does not express any prioritisation. The Research Concept itself is adjusted every 4 years, but measures and research goals in hazard prevention are only adjusted if knowledge gaps are identified or existing ones are closed.

**Research themes in hazard prevention 2021 – 2024[1]**

**1. Know hazards and risks comprehensively**

- Generation of basic scientific knowledge for hazard processes like e.g. lateral erosion of watercourses, hillslope debris flows, rockfall, snow gliding avalanches, impulse waves in Swiss lakes triggered by mass movements or earthquakes (tsunamis on lakeshores) and of the influence of climate change on such processes

- Development of methods to record natural hazard processes, e.g. sediment and driftwood transport, to register
indirect damages due to natural hazard events and development of methods to assess future hazards and extreme events due to climate change. A good example for this topic is the recently completed WoodFlow research project (www.woodflow.ch), which improves the understanding of the processes governing large wood dynamics in watercourses and provides practitioners with suitable tools to help assess and manage large wood related hazards.

- Development of a methodology for risk overviews at different spatial scales

- Investigation of the impact of climate change on the risk landscape in Switzerland and analysis of combinations and concatenations of different processes

- Elaboration of fundamentals to quantify the vulnerability and the risk for infrastructural systems in the event of earthquakes and gravitational natural hazards

**2. Identify events at an early stage**

- Investigation of precipitation thresholds and analysis of the disposition to slope processes in-depth. The FOEN is currently developing a warning system for landslides and hillslope debris flows. These disposition warnings provide indications at various warning levels as to the areas and probability of slopes becoming unstable due to the current

water saturation. The monitoring of landslide areas is also being intensified. With the InSAR (satellite-based radar interferometry) method, it is possible to monitor mass movements in the alpine region and to detect new movements.

- Improvement of extreme value statistics
- Improvement of the forecasting and early detection of natural hazards

### 3. Plan measures integrated and to be robust

- Development of methods for evaluating options for action within the integrated risk management
- Development of new instruments for risk-based land use and their practical implementation in spatial planning
- Analysis of ecological aspects in the implementation of protective measures; addressing of dichotomy of protective function of the forest vs the forest as a habitat for game animals. The new research program River Engineering and Ecology (https://www.rivermanagement.ch) is a good example for this topic. It aims to develop scientific foundations for answering current practical questions in river management (flood protection and renaturation of water bodies) and to prepare outreach products for knowledge transfer.
- Development of methods to assess existing protective structures and to assess robust protective systems; development of models for dynamic loads on structures

### 4. Search the risk dialogue and observe the impact on society

- Investigation of the social science component in risk perception and communication; conduction of research into the acceptance of measures
- Further development of cost-benefit analysis in integrated risk management; development of methods for quantifying the indirect economic damage after earthquakes and gravitational natural hazards
- Integration of organisational and human factors in risk identification and assessment

### 5. Mitigate the seismic risks

- Development of methods for assessing and retrofitting cultural-historical buildings
- Development of methods for the consideration of seismically triggered secondary hazards in hazard and risk analyses
- Optimisation of earthquake safety requirements in building codes
- Development of new technologies for damage and residual load-bearing capacity assessments of buildings after earthquakes

Established cooperation between Swiss research institutions and foreign partners exist. The FOEN maintains a continuous exchange of knowledge with administrations in other countries, e.g. in the INTERPRAEVENT and in the context of the Platform on Natural Hazards of the Alpine Convention (PLANALP) and with the EU in the framework of the EU Strategy for the Alpine Region (EUSALP). Switzerland cooperates, among others, with neighbouring countries with which it shares lakes (e.g. Lake Maggiore, Lake Geneva) or where rivers cross both countries (e.g. International Commission for the Protection of

the Rhine, ICPR). Switzerland's experience in dealing with natural hazards and its approach to reducing risks it also brings to bear in international committees and conferences, such as the Global Platform for Disaster Risk Reduction. The Swiss Agency for Development and Cooperation (SDC) is also able to call on the FOEN's experience and expertise in prevention and protection projects.

**What concerns everyone can only be resolved by everyone.** In Switzerland, the municipalities and cantons are primarily responsible for protection against natural hazards. The Confederation assumes its strategic leadership role and supports the cantons financially and technically. Other important tasks managing the risks from natural hazards are assumed by the insurance companies in accordance with their legal mandate. They provide financial cover for potential damage. By promoting preventive measures and providing information and advice to house owners, they make a significant contribution to integrated

risk management. With the help of their standards, associations provide a basis that serves as a planning aid for construction that is suitable for natural hazards. Private actors and those directly affected are required to ensure protection against natural hazards as far as possible in accordance with the principle of Art. 6 of the Federal Constitution, according to which each person assumes responsibility for him or herself. Private actors and those directly affected must be encouraged to take the initiative at least in the sense that they do not expose themselves to natural hazards, protect themselves from them or try to minimise

their effects themselves. Incentives and cooperative forms can be considered for this purpose, e.g. insurance premium incentives for property protection measures, inclusion of insurance companies in the preliminary review of communal land use planning or in building permit procedures.

However, monitoring, warning systems, protective structures, emergency services and insurances alone are not enough to prevent damage. Everyone, from house owners to tenants, from the national railway operators to power providers, from

hoteliers to freight carriers, can be affected by natural hazards, anywhere in the country. Therefore, they have to contribute as well, by object protection measures on houses and infrastructures or by hazard-appropriate behaviour in the case of an event. As the famous Swiss author Friedrich Dürrenmatt once wrote, "What concerns everyone can only be resolved by everyone." Only if all actors concerned take on their responsibility, Switzerland can achieve and maintain an adequate safety for people, property and natural resources throughout the country.


*Author contributions.* The authors jointly wrote the manuscript.

*Competing interests.* The authors declare that they have no conflict of interest.

*Data availability.* No data sets were used in this article.

*Disclamer.* Publisher's note: Copernicus Publications remains neutral with regard to jurisdictional claims in published maps and institutional affiliations.

*Special issue statement.* This article is part of the special issue "Perspectives on challenges and step changes for addressing natural hazards". It is not associated with a conference.

*Acknowledgements.* We would like to thank the reviewers and the editor who helped us to significantly improve the manuscript.

*Review statement.* This paper was edited by XXX and reviewed by two anonymous referees.

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
