# Peer review of "Invited Perspectives: "Small country, big challenges – Switzerland's hazard prevention research""

_Natural Hazards and Earth System Sciences, 2021_

## Author Response (AR1)

**Point-by-point reply to the comments to NHESS 2021-190 Invited Perspectives: "Small country, big challenges – Switzerland's hazard prevention research"**

Dorothea Wabbels[1], Gian Reto Bezzola[1]

[1]Federal Office For The Environment (FOEN), Hazard Prevention Division, 3003 Berne, Switzerland

*Correspondence to*: Dorothea Wabbels (dorothea.wabbels@bafu.admin.ch)

**Review No. 1**

- *The list of research topics provides a rough overview. However, this part of the paper would gain a lot if 3-4 of these topics were dealt with more soundly and in more detail. Thus, I suggest that you pick 3-4 topics as examples and describe the state of the art of this research topic and what innovations are expected via this research plan within the next 4 years. Clear examples of ongoing research projects to improve knowledge in these areas would be very helpful.*
  → **We do agree and added some examples. The research concept itself is adjusted every 4 years, but our measures and research goals are only adjusted if new knowledge gaps are identified or existing ones are closed. We added this information in the article.**

- *The paragraph "What concerns everyone can only be resolved by everyone" is too general to provide any interesting information. Please describe the integrated risk management approach in Switzerland in more detail. How is it organized? How well does it work? What are remaining challenges and ideas how to solve current problems. The concept, that all actors need to work together to achieve an effective risk management is good, but what is done to achieve this vision? For instance, in Germany, according to § 5 of the German Federal Water Resource Act that was enacted in 2009, every person that could be affected by a flood is obliged to undertake appropriate actions that are reasonable and within one's means to reduce flood impacts and damage. Does a similar law exist in Switzerland? What incentives are provided to motivate private precaution?*
  → **We added a short description of the integrated risk management approach. Going into detail answering all the questions concerning IRM would be beyond the topic of this article (research). In Switzerland, the municipalities and cantons are primarily responsible for protection against natural hazards. The Confederation assumes its strategic leadership role and supports the cantons financially and technically. Other important tasks for protection against natural hazards are assumed by the insurance companies in accordance with their legal mandate. They provide financial cover for potential damage. By promoting preventive measures and providing information and advice to customers, they make a significant contribution to protection against natural hazards. With the help of their standards, professional associations provide a basis that serves as a planning aid for construction that is suitable for natural hazards. Private actors and those directly affected are required to ensure protection against natural hazards as far as possible in accordance with the principle of Art. 6 of the Federal Constitution, according to which each person assumes responsibility for him or herself and contributes to the accomplishment of tasks in the state and society to the best of his/her ability. Incentives and cooperative forms can be considered for this purpose, e.g. insurance premium incentives for property protection measures, inclusion of insurance companies in the preliminary review of communal land use planning or in building permit procedures. Research takes place about nudging - the aim is to find out which "nudges" the insurer has to use to persuade the relevant actors in the loss and reassessment situations to increase the protection of their building** (Projekte | Vereinigung Kantonaler Gebäudeversicherungen (VKG)**). As this is not part of FOEN's research concept, we will not add this information in the article.**

- *Line 17: No "seashore" in Switzerland. Better "along rivers and lakes".*
  → **Yes, we do agree and changed it to "along rivers and lakes".**

- *Fig. 1 Do you have the right to publish this figure? Please provide evidence of this, e.g. written consent from WSL.*
  → **Yes, we do have the rights. Since 1972 the Swiss Federal Research Institute WSL has been systematically collecting (based on newspapers) and analysed this damage on behalf of us, the Federal Office for the Environment FOEN (**Swiss flood and landslide damage database - WSL**). We added this information.**

**Review No. 2**

- *I wonder what the naming of priorities in the perspective (and in the research concept) actually means. Is this a list of equally important research fields? As I have not seen a prioritization between them, the concept might be a "first come first served" concept. Alternatively, is the idea that all areas are equally covered?*
→**The order does not express any prioritisation. We added this information in the article.**

- *While it appears that the research priorities are subject to regular change ("concept 2021-2024"), the challenges named in the first part appear to be rather persistent. It should be mentioned, if possible, what basically determines the longer term changes in the research concept.*
→**The Research Concept itself is adjusted every 4 years, but measures and research goals in hazard prevention are only adjusted if knowledge gaps are identified or existing ones are closed. We added this information in the article.**

- *It appears that the FOEN's perspectives is be limited to hazards relevant to Switzerland. As basic tsunami research is mentioned, it might be worthwhile to mention in how far work on non-Swiss hazards are part of the FOEN perspective or left to the consideration of other agencies.*
→**Our applied tsunami research aims for hazard assessment along Swiss lakeshores. Established cooperations between Swiss research institutions and foreign partners exist. The FOEN maintains a continuous exchange of knowledge with administrations in other countries, e.g. in the INTERPRAEVENT and in the context of the International Commission for the Protection of the Rhine ICPR, Platform on Natural Hazards of the Alpine Convention (PLANALP) and with the EU in the framework of the EU Strategy for the Alpine Region (EUSALP). We added this information in the article.**

- *How is the final pledge for a role of everyone supported? With respect to a research perspective, could it be, for example, that sociological research is needed to improve the readiness of "everyone" to improve their personal responsibility in the field of natural hazards?*
→ **Natural hazard prevention is a cross-sectional task, encompassing land use, spatial planning, water protection, environmental and civil protection by the government, cantons and communes, the private sector as well as insurance companies and individuals. These actors are involved in the process as a joint task, in accordance with the principle of subsidiarity. Investigation of the social science component in risk perception and communication is part of the research concept. Voters in Switzerland decide on political issues on the national, cantonal and municipal level up to four times a year. Votes are e.g. held on major expenditures, as e.g. for natural hazard protection projects. Thus, there is a regular influence of the citizens on expenditures through this approval process. We added this information in the article.**

- *(line 15) The storm Lothar could be combined with the storm Martin, as both form a storm sequence affecting the area ( see, e.g., https://doi.org/10.5194/nhess-21-279-2021 ).*
→ **The storm Martin did not cause large damages in Switzerland, but we added Burglind 2018, the 2nd strongest storm after Lothar in Switzerland.**

- *In Figure 1, it is difficult to distinguish the different shades of grey. A coloured layout would improve readability significantly.*
→ **We updated and modified the figure to make it readable.**

- *The references given appear to be links, but these links are not explicitly included. If possible, DOIs should be given, and regular publication information if possible*
→ **no DOI available (grey literature)**

---

## Author Response (AR2)

- 40 Small country, big challenges. Switzerland has a long tradition concerning fundamental research as well as practice- and implementation-oriented research in hazard management and there is an important number of research institutions being active in this domain. The findings of these research activities will further improve the management of natural hazards and, taking into account environmental aspects, climate change and settlement densification. The «Research Concept Environment » (FOEN 2020b) describes the implementation of environmental policies in 18 research areas and thus the priority fields of
- 45 research from the perspective of FOEN in the near- and future-term. FOEN's focus on central research themes in hazard prevention of gravitational and tectonical hazards for 2021 – 2024 is listed below. The order does not express any prioritisation. The Research Concept itself is adjusted every 4 years, but measures and research goals in hazard prevention are only adjusted if knowledge gaps are identified or existing ones are closed.

**Research themes in hazard prevention 2021 - 20241**

**50 1. Know hazards and risks comprehensively**

- Generation of basic scientific knowledge for hazard processes like e.g. lateral erosion of watercourses, hillslope debris flows, rockfall, snow gliding avalanches, impulse waves in Swiss lakes triggered by mass movements or earthquakes (tsunamis on lake shores) and of the influence of climate change on such processes
- Development of methods to record natural hazard processes, e.g. sediment and driftwood transport, to register indirect damages due to natural hazard events and development of methods to assess future hazards and extreme events due to climate change. A good example for this topic is the recently completed WoodFlow research project (www.woodflow.ch), which improves the understanding of the processes governing large wood dynamics in watercourses and provides practitioners with suitable tools to help assess and manage large wood related hazards.
  - Development of a methodology for risk overviews at different spatial scales
- Investigation of the impact of climate change on the risk landscape in Switzerland and analysis of combinations and concatenations of different processes
  - Elaboration of fundamentals to quantify the vulnerability and the risk for infrastructural systems in the event of earthquakes and gravitational natural hazards

**2. Identify events at an early stage**

• Investigation of precipitation thresholds and analysis of the disposition to slope processes in-depth. The FOEN is currently developing a warning system for landslides and hillslope debris flows. These disposition warnings provide indications at various warning levels as to the areas and probability of slopes becoming unstable due to the current

**55**

60